# On the Non-Abelian U-Duality of 11D Backgrounds

Edvard T. Musaev [1,2]

1    Moscow Institute of Physics and Technology, Institutskii per. 9, 141700 Dolgoprudny, Russia;
     musaev.et@phystech.edu
2    Institute of Physics, Kazan Federal University, Kremlevskaya 16a, 420111 Kazan, Russia

**Abstract:** In this work, we generalise the procedure of the non-abelian T-duality based on a B-shift and a sequence of formal abelian T-dualities in non-isometric directions to 11-dimensional backgrounds. This consists of a C-shift followed by either a formal (abelian) U-duality transformation or taking an IIB section. By construction, this is a solution generating transformation. We investigate the restrictions and applicability of the procedure and find that it can provide supergravity solutions for the SL(5) exceptional Drinfeld algebra only when the isometry algebra of the sigma-model target space decomposes into a direct sum. This is consistent with examples known in the literature.

**Keywords:** supergravity; dualities; M-theory

# Contents

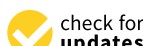



## 1. Introduction

String theory is known to respect a rich set of various symmetries, of special interest among which are those that transform target space–time keeping the same physics. The most well-known example of such duality symmetries is the perturbative T-duality symmetry of Type II string theory, which acts along toroidal directions of the target space according to the so-called Buscher rules [1,2]. The procedure for recovering background fields' transformations from the string partition function is well known. One starts with the string partition function defined by the action $S_0[\theta]$ symmetric under global $\theta \to \theta + \alpha$ with $\theta$ corresponding to a circular direction. The symmetry is then gauged by introducing a 1-form field $d\theta \to D\theta = d\theta + A$ and the corresponding Lagrange term $\tilde{\theta}F$ with $F = dA$ to keep the 1-form pure gauge. The resulting partition function defined by the action $S_1[\theta, A, \tilde{\theta}]$ can then be reduced to the initial one, integrating out $\tilde{\theta}$, which sets $A = d\alpha$.

Alternatively, integrating out the 1-form field $A$, one obtains the standard string action $S_2[\tilde{\theta}]$, however defined on a different background related to the initial one by Buscher rules. The scalar field $\theta(\sigma, \tau)$ gets replaced by the field $\tilde{\theta}(\sigma, \tau)$ representing dual string coordinates corresponding to winding modes [3,4]. Transformation of the dilaton ensures that the measure in the partition function is invariant at one loop. One can be more general and consider backgrounds of the form $M \times \mathbb{T}^d$ in which case the T-duality group will be $O(d, d; \mathbb{Z})$. World-volume scalar fields that do not transform under the duality are referred to as spectator fields.

A natural question is whether one may consider backgrounds with isometries represented by more complicated groups than the abelian $U(1)^d$, say a sphere or a non-abelian group manifold. The answer is positive, and the corresponding dualisation procedure was considered in [5]. Essentially, the non-abelian T-duality (NATD) of the string partition function goes along the same lines as the abelian one. The difference comes from the more involved definition of the field strength $F = dA + [A, A]$, which is now an element of the corresponding non-abelian algebra, and hence, the Lagrange term reads $\text{Tr}[\tilde{\theta}F]$. Hence, one dualises the whole set of group coordinates, basically replacing left-invariant 1-forms $\sigma^a$ with dual forms $d\tilde{\theta}_a$. The original procedure for NS-NS fields was complemented by transformation rules for R-R fields in [6,7]. An explicit canonical formulation of non-abelian T-duality for the principal sigma-model was provided in [8]. In the work [7], the procedure was extended to coset space geometries $G/H$ based on fixing the gauge degrees of freedom corresponding to the action of the subgroup $H$. One should be careful here with global issues of non-abelian T-dual backgrounds related to determining the range of dual coordinates. Recall that the case of the abelian T-duality topology of the vector potential $A$ requires the dual coordinate to be periodic, and hence, the $U(1)$ isometry is preserved. Similarly here, one looks at 2-cycles and topological properties of the field $B$ such that its integral properly rescaled takes values in the interval $[0, 1]$. The procedure for determining the ranges was suggested in [9] based on earlier works [10–12].

In contrast to abelian T-duality, its non-abelian generalisation does not preserve isometries of the original background (in the usual sense) and, hence, has much in common with deformations of supergravity backgrounds. In particular, NATD techniques have been widely used to generate new supergravity backgrounds interesting from the point of view of holography, and in [13], some explicit examples of such relations were provided. This breaking of the initial background isometries by a non-abelian T-duality transformation is in severe contrast with the mechanics of the standard abelian T-duality transformations, where the preservation of isometries allows performing T-duality twice, making it an involutive symmetry. For a way out of this problem and to define an inverse for an NATD transformation, one looks at the Noether currents of the two-dimensional string sigma-model and their Bianchi identities. Starting with the sigma-model on a background with isometry algebra defined by structure constants $f_{ab}{}^c$, one is able to construct conserved Noether currents $J_a$ that satisfy

$$dJ_a = 0. \tag{1}$$

Non-abelian T-dualising along the isometry directions, one ends up with the sigma-model on a background with no initial isometries, which however still allows defining Noether currents $\tilde{J}^a$ that are not conserved and satisfy [14]

$$d\tilde{J}^a = f_{bc}{}^a \tilde{J}^b \wedge \tilde{J}^c. \tag{2}$$

In principle, one is able to start even with backgrounds with no isometry in the standard sense with Noether currents that satisfy

$$dJ_a = \tilde{f}_a{}^{bc} J_b \wedge J_c. \tag{3}$$

For the duality between the backgrounds to work, the algebras $\mathfrak{g}$ and $\tilde{\mathfrak{g}}$ defined by the structure constants $f_{ab}{}^c$ and $\tilde{f}_a{}^{bc}$ must form the so-called Drinfeld double $\mathcal{D}$. This is a $2d$-dimensional Lie algebra spanned by generators $\{T_a, \tilde{T}^a\}$ with an $O(d, d)$ invariant metric

$\eta$ defined such that the only non-vanishing component is $\eta(T_a, \tilde{T}^b) = \delta_a{}^b$. Generators $\{T_a\}$ define the isotropic subalgebra $\mathfrak{g}$, while its complement $\tilde{\mathfrak{g}}$ is spanned by $\{\tilde{T}^a\}$. In addition, one has the co-algebra structure. In principle, such a choice of generators for a given Drinfeld double Lie algebra $\mathcal{D}$ might not be unique, and hence, it is convenient to think of it in terms of Manin triples $(\mathcal{D}, \mathfrak{g}, \tilde{\mathfrak{g}})$. Such algebraic construction allows reversing the NATD transformation applying a Poisson–Lie T-duality (PLTD) transformation, which basically means solving consistency constraints for the Drinfeld double and constructing a background that realises the chosen geometric subalgebra $\mathfrak{g}$. More details on relations between Poisson–Lie and non-abelian dualities can be found in the original works [15,16] and in the review papers [17–19]. For developments from the generalised geometry point of view, one may refer to [20–23]. Explicit examples of backgrounds resulting from PLTD and/or NATD can be found in [7,24–28]. The representation of Yang–Baxter bi-vector deformations as a B-shift followed by an NATD transformation was considered in [29].

To some extent, the above constructions generalise to M-theory in the sense of membrane dynamics and 11-dimensional supergravity. The notion of the Drinfeld double (Manin triple) were generalised to the so-called exceptional Drinfeld algebra (EDA) in the series of works [30,31]. The complement $\tilde{\mathfrak{g}}$ of the isometry algebra $\mathfrak{g}$ inside an exceptional Drinfeld algebra is defined via tri-algebra structure constants $\tilde{f}_a{}^{bcd}$. From the membrane point of view, non-abelian U-duality was addressed in [32], where in particular, an analogue of Bianchi identities for currents of the two-dimensional sigma-model were derived and implemented in the SL(5) exceptional field theory (ExFT). The current algebra of [32] is consistent with the 3-algebra structure of [30,31]. Finally, certain explicit results for non-abelian U-dualised backgrounds and their relation to non-abelian T-duality were presented recently in [33].

As we discuss below, the procedure of non-abelian U-duality (and more generally, Nambu–Lie U-duality) based on the algebraic approach of [30,31] includes the pretty non-trivial step of searching for a possible alternative realisation of a given exceptional Drinfeld algebra. Technically, one has to find such an $\mathrm{E}_{d(d)}$ element whose action on generators of a given EDA gives also an EDA. In this work, based on the results of [29] on non-abelian T-duality, we (i) suggest to include outer automorphisms of $\mathrm{E}_{d(d)}$ in the set of allowed transformations and (ii) show that this always gives a solution to 11-dimensional supergravity equations. Hence, this is a slight generalisation of the prescription of [30,31] and its translation to the language of field theory. The suggested procedure provides transformation rules of non-abelian U-duality that are guaranteed to always give a solution of 11-dimensional supergravity equations by construction. Explicit examples of non-abelian U-duals based on the suggested procedure were provided in [34].

The text is structured as follows. In Section 2, we review the NATD procedure as an $\mathrm{O}(d,d)$ rotation for group manifolds. As an explicit example, the Bianchi II space–time with vanishing dilaton is considered. In Section 3, we generalise the approach to non-abelian U-duality transformations of 11-dimensional backgrounds. In Section 4, we analyse the suggested procedure for ExFT's based on U-duality groups SL(5) and SO(5,5) and derive conditions upon which a solution of 11-dimensional supergravity can be generated.

## 2. Non-Abelian T-Duality

### 2.1. Sigma-Model Perspective

Non-abelian T-duality transformations generalise standard T-duality Buscher rules and can be written in a very similar form [29]. The case of our interest here is represented by backgrounds of the form $M \times G$ where $G$ is a group manifold; however, the sigma-model procedure can be generalised to coset spaces. To set up the notations, we briefly discuss the procedure of [29] here. One starts with a two-dimensional sigma-model action of the form

$$S = T \int_\Sigma \left( \frac{1}{2} E^{\hat{\alpha}} \wedge *E^{\hat{\beta}} \eta_{\hat{\alpha}\hat{\beta}} + B \right), \tag{4}$$

where the vielbein 1-form $E^{\hat{\alpha}}$ is defined as usual as

$$
\begin{aligned}
E^{\hat{\alpha}} &= (g^{-1}dg)^a E_a{}^{\hat{\alpha}} + dx^\mu E_\mu{}^{\hat{\alpha}}, \quad g \in G, \\
g^{-1}dg &= \sigma_m{}^a dy^m T_a.
\end{aligned}
\tag{5}
$$

Here and in what follows, small Greek indices $\mu, \nu$ label external directions, which are not extended/doubled, small Latin indices $a, b, \cdots = 1, \ldots, \dim G$ from the beginning of the alphabet label generators of Lie algebra $\mathfrak{g}$ of the group manifold $G$, and small Latin indices from the middle of the alphabet $m, n, \cdots = 1, \ldots, \dim G$ label coordinates $y^m$ on the group manifold. Functions $\sigma_m{}^a$ represent components of left-invariant 1-forms on the group manifold, and $T_a$ form a basis of the corresponding Lie algebra $\mathfrak{g}$. Isometry transformations act on the group manifold from the left as

$$
g \to ug, \quad u \in G.
\tag{6}
$$

Unpacking these notations, one may write for the first term in the sigma-model action

$$
\begin{aligned}
E^{\hat{\alpha}} \wedge *E^{\hat{\beta}} \eta_{\hat{\alpha}\hat{\beta}} &= (g^{-1}dg)^a \wedge *(g^{-1}dg)^b G_{ab} \\
&\quad + 2(g^{-1}dg)^a \wedge *dx^\mu G_{a\mu} + dx^\mu dx^\nu G_{\mu\nu},
\end{aligned}
\tag{7}
$$

where one defines metric components:

$$
\begin{aligned}
G_{\mu\nu} &= E_\mu{}^{\hat{\alpha}} E_\nu{}^{\hat{\beta}} \eta_{\hat{\alpha}\hat{\beta}}, \\
G_{mn} &= \sigma_m{}^a \sigma_n{}^b G_{ab} = \sigma_m{}^a \sigma_n{}^b E_a{}^{\hat{\alpha}} E_b{}^{\hat{\beta}} \eta_{\hat{\alpha}\hat{\beta}}, \\
G_{m\mu} &= \sigma_m{}^a G_{a\mu} = \sigma_m{}^a E_a{}^{\hat{\alpha}} E_\mu{}^{\hat{\beta}} \eta_{\hat{\alpha}\hat{\beta}}.
\end{aligned}
\tag{8}
$$

The 2-form Kalb–Ramond field $B$ is defined as usual as a pullback of the corresponding target space–time 2-form field:

$$
\begin{aligned}
B &= (g^{-1}dg)^a \wedge (g^{-1}dg)^b B_{ab} \\
&\quad + 2(g^{-1}dg)^a \wedge dx^\mu B_{a\mu} + dx^\mu \wedge dx^\nu B_{\mu\nu}.
\end{aligned}
\tag{9}
$$

The fields $G_{ab}, B_{ab}$ are usually referred to as the undressed fields since these are free of dependence on group coordinates $y^m$, which has all been left in the 1-forms $\sigma^a$.

The procedure of the NATD of the sigma-model action then proceeds with replacing $(g^{-1}dg)^a \to A^a$ and adding a Lagrange multiplier $\tilde{y}_a F^a$. Performing integration over $\tilde{y}_a$, one recovers the initial action, while integrating over $A^a$, one turns to the dual action, which now has no dependence on $y^m$ since the 1-forms $\sigma^a$ are no longer present. Instead, a dependence on $\tilde{y}_a$ enters the dual background originating from

$$
F^a = 2dA^a - f_{bc}{}^a A^b \wedge A^c,
\tag{10}
$$

where $f_{ab}{}^c$ encode the structure constants of $\mathfrak{g}$.

This procedure can be summarised nicely by presenting a generalisation of Buscher rules, explicitly providing dual background fields. For that, one defines a matrix:

$$
N_{ab} = G_{ab} - B_{ab} + \tilde{y}_c f_{ab}{}^c,
\tag{11}
$$

along with its inverse $N^{ac} N_{cb} = \delta^a{}_b$. The transformation rules are then written as follows:

$$G'_{\mu\nu} = G_{\mu\nu} - (G-B)_{a(\mu}N^{ab}(G+B)_{\nu)b}$$

$$G'_{\mu a} = \frac{1}{2}(G-B)_{\mu b}N^{ba} - \frac{1}{2}N^{ab}(G-B)_{b\mu}$$

$$G'_{ab} = N^{(ab)}$$

$$B'_{\mu\nu} = B_{\mu\nu} + (G-B)_{a[\mu}N^{ab}(G+B)_{\nu]b} \qquad (12)$$

$$B'_{\mu a} = -\frac{1}{2}(G-B)_{\mu b}N^{ba} - \frac{1}{2}N^{ab}(G-B)_{b\mu}$$

$$B'_{ab} = -N^{[ab]}$$

These were shown in [22] to be upliftable to the double field theory (DFT) formalism where the transformation of the fields becomes an $O(d,d)$ matrix with $d = \dim G$, as expected. Crucial here is that the 1-forms $\sigma^a$ are replaced by dual 1-forms $d\tilde{y}_a$, constructed of dual coordinates. For abelian T-duality, one would have $dy^a \to d\tilde{y}_a$, which translates to the standard prescription $y^a \to \tilde{y}_a$.

### 2.2. Double Field Theory Perspective

The non-abelian T-duality transformation of a 10-dimensional (group manifold) background as described above is known to be equivalent to a sequence of a B-shift and T-duality transformations, equivalently $O(d,d)$ reflections [35]. The procedure can be generalised to coset spaces as well, where one chooses $d$ Killing vectors in a $d$-dimensional space and takes basically the same steps. Crucial is that the symmetry group acts without isotropy. In the present text, we focus on the case of group manifolds to illustrate the procedure and to make further analysis of its restrictions simpler. Given the results of [35], generalisation to coset spaces must be straightforward.

In this section, we take the perspective of double field theory, which provides a convenient framework for dealing with dual coordinate dependence [36–39]. Double field theory is a field theory of the metric $G_{mn}$, the 2-form $B_{mn}$, and the dilaton $\phi$ combined into irreducible representations of the global T-duality group $O(d,d)$. Fields are allowed to depend on a doubled set of coordinates $\mathbb{X}^M = (x^m, \tilde{x}_m)$ with $M = 1, \ldots, 2d$ labelling the vector representation of $O(d,d)$. On such a doubled space (space–time, if $d = 10$), one defines transformations of a generalised vector field $V^M$ using the so-called generalised Lie derivative:

$$\delta_\Lambda V^M = \mathcal{L}_\Lambda V^M = \Lambda^N \partial_N V^M - (\partial_N \Lambda^M - \partial^M \Lambda_N) V^N, \qquad (13)$$

where the indices are raised and lowered by the invariant metric $\eta_{MN}$ of $O(d,d)$ taken to be

$$\eta_{MN} = \begin{bmatrix} 0 & 1 \\ 1 & 0 \end{bmatrix}. \qquad (14)$$

The first term in (13) is the standard shift term, and the second is a local $O(d,d)$ rotation of the generalised vector. Such defined generalised Lie derivatives form a closed algebra under the standard commutator:

$$[\delta_{\Lambda_1}, \delta_{\Lambda_2}]V^M = \delta_{[\Lambda_1, \Lambda_2]}V^M,$$

$$[\Lambda_1, \Lambda_2] = \frac{1}{2}\mathcal{L}_{\Lambda_1}\Lambda_2 - \frac{1}{2}\mathcal{L}_{\Lambda_2}\Lambda_1 \qquad (15)$$

only if the so-called section condition is satisfied. This condition is a special constraint on functions defined on such a doubled space and is usually schematically stated as

$$\eta^{MN}\partial_M \bullet \partial_N \bullet = 0, \qquad (16)$$

where the bullets stand for any fields and their combinations. This isotropic condition simply states that all fields are allowed to depend on (the same) set of $d$ coordinates out of $(x^m, \tilde{x}_m)$. The degrees of freedom of supergravity are packed into the so-called generalised metric $\mathcal{H}_{MN}$ and the invariant dilaton $d$ defined as follows

$$\mathcal{H}_{MN} = \begin{bmatrix} G_{mn} - B_{mk}B^k{}_n & B_m{}^n \\ B_n{}^m & G^{mn} \end{bmatrix}, \quad d = \phi - \frac{1}{4}\log G, \tag{17}$$

where $G = \det G_{mn}$. The dynamics of these fields is encoded in the action:

$$S = \int dx d\tilde{x} e^{-2d} \left( \frac{1}{8}\mathcal{H}^{MN}\partial_M \mathcal{H}^{KL}\partial_N \mathcal{H}_{KL} - \frac{1}{2}\mathcal{H}^{KL}\partial_L \mathcal{H}^{MN}\partial_N \mathcal{H}_{KM} - \right.$$
$$\left. -2\partial_M d\partial_N \mathcal{H}^{MN} + 4\mathcal{H}^{MN}\partial_M d\partial_N d \right). \tag{18}$$

Since the integral of such constrained fields over the doubled space is not defined in the strict mathematical sense, the above must be understood as a condensed form of writing the corresponding field equations.

The section condition restricts fields of the theory to depend at most on half of the full 2d-dimensional set of coordinates; however, it does not tell on which. This is additional information that has to be put in by hand, and different choices are related by $O(d,d)$ reflections. The standard choice is to make fields depend purely on the geometric coordinates $x^m$, which makes the DFT action simply the standard supergravity action. Alternatively, one may solve the condition such that the fields depend, say, on nine coordinates $x^0, \ldots, x^8$ and on the dual coordinate $\tilde{x}_9$. Say, for NS 5-branes, this allows localising KK5-monopole and more exotic branes [40,41]. A very particular example of such backgrounds is when all fields depend on $x^0, \ldots, x^8$ and the dilaton additionally linearly depends on $\tilde{x}_9$. Since, the dilaton enters DFT equations only via its derivatives, the dependence on the dual coordinate manifests itself only through a slight modification of the supergravity equations. The resulting theory is usually referred to as generalised supergravity [42–44].

Given all that, the main feature of DFT relevant to the present discussion is that it allows more general backgrounds than that of the ordinary supergravity, i.e., those that depend on dual coordinates and respect the section constraint. Of course, these are related by a formal (abelian) T-duality transformation along non-isometric directions to normal supergravity backgrounds, as the localised KK5 is a formal T-dual of the NS5-brane (without smearing). This allows easily checking that the B-shift + T-dualities procedure always gives a solution of supergravity backgrounds and can be easily generalised to exceptional field theories.

One starts by noticing that, to generalise the NATD transformation rules written in the form (12) to 11d backgrounds, it is convenient to rewrite these in terms of an $O(d,d)$ rotation of a DFT background. Following [35], the algorithm is as follows:

- Undress the background fields;
- Perform a B-shift $B_{ab} \to B_{ab} + \tilde{y}_c f_{ab}{}^c$, with $\tilde{y}_a$ understood as the coordinates dual to 1-forms $\sigma^a = \sigma_m{}^a dy^m$;
- Perform formal abelian T-dualities along all directions of the group manifold to turn $\tilde{y}_a$ into geometric coordinates.

Schematically, the procedure is depicted in Figure 1.

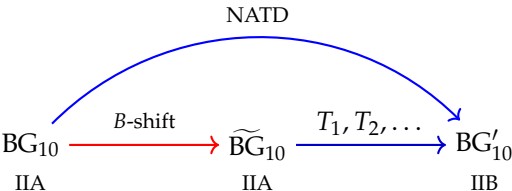

**Figure 1.** Relationship between backgrounds upon non-abelian T-duality. $T_i$ denotes the usual T-duality along the $i$'th direction.

For further reference and to set up the notations, let us consider the procedure in more detail. The first step splits the coordinate dependence to external coordinates $x^\mu$ and group manifold coordinates $y^m$ hidden in 1-forms $\sigma_m{}^a$:

$$
\begin{aligned}
G_{mn}(x,y) &= \sigma_m{}^a(y)\sigma_n{}^b(y)G_{ab}(x), \\
B_{mn}(x,y) &= \sigma_m{}^a(y)\sigma_n{}^b(y)B_{ab}(x).
\end{aligned}
\tag{19}
$$

Turning to undressed fields, which do not depend on the coordinates $y^m$, one ensures that the B-shift respects the section constraint. To perform further abelian T-duality transformation, one has to consider a doubled set of 1-forms $(\sigma^a, d\tilde{y}_a)$, instead of coordinates. The transformation of generalized metric $\mathcal{H}_{AB}$ constructed from the undressed fields $(G_{ab}, B_{ab})$ can be conveniently read off the so-called DFT pseudo-interval. Let us first illustrate this in a more standard case.

Start with the following formal expression:

$$
\begin{aligned}
ds^2 &= \mathcal{H}_{MN}d\mathbb{X}^M d\mathbb{X}^N \\
&= \mathcal{H}_{mn}dy^m dy^n + 2\mathcal{H}_m{}^n dy^m d\tilde{y}_n + \mathcal{H}^{mn}d\tilde{y}_m d\tilde{y}_n,
\end{aligned}
\tag{20}
$$

that is neither invariant under generalised coordinate transformations, nor represents any distance measurement. Rather, it serves as a convenient form of encoding the roles of the components of the generalised metric, distinguishing the metric and the B-field. Here and in what follows, capital Latin indices $M, N, \ldots$ label the directions of the extended space and in case of DFT, run $1, \ldots, 2\dim G$. Assuming that $y^m$ and $\tilde{y}_m$ are the geometric and dual coordinates, respectively, one thus fixes $\mathcal{H}^{mn} = g^{mn}$. To perform a T-duality transformation, one keeps the pseudo-interval the same, switching instead the roles of the coordinates. Say $y^1$ now becomes a dual, while $\tilde{y}_1$ becomes geometric, meaning that $\tilde{y}_1$ is the coordinate used to measure the distance in the ordinary space. This implies that $\mathcal{H}_{11} = \tilde{g}^{11}$ is now a component of the transformed metric (not the inverse). This procedure was employed to generate exotic brane solutions and to unify them into a single DFT/ExFT solution in [40,41,45,46].

For the case in question, it is natural to write the pseudo-interval in terms of the undressed fields as

$$
\begin{aligned}
ds^2 &= \mathcal{H}_{MN}d\mathbb{X}^M d\mathbb{X}^N \\
&= \mathcal{H}_{ab}\sigma^a\sigma^b + 2\mathcal{H}_a{}^b\sigma^a d\tilde{y}_b + \mathcal{H}^{ab}d\tilde{y}_a d\tilde{y}_b,
\end{aligned}
\tag{21}
$$

where the dependence on $y^m$ was recollected into the 1-forms $\sigma^a = \sigma^a{}_m dx^m$. For now, the dual coordinates are still represented by exact forms $d\tilde{y}_a$.

Let us show that the procedure described above is guaranteed to end up with a solution of supergravity equations if started with a solution. Indeed, let us start for simplicity with a background that depends purely on group manifold coordinates, i.e., $G_{ab} = \text{const}$, $B_{ab} = \text{const}$. Hence, one may encode the invariant 1-forms in a generalised vielbein $E_M{}^A$ and the constant metric and B-field in the constant «flat» generalised metric $\mathcal{H}_{AB}$:

$$
\mathcal{H}_{MN} = E_M{}^A E_N{}^B \mathcal{H}_{AB}, \quad \mathcal{H}_{AB} = \begin{bmatrix} G_{ab} - B_{ae}B^e{}_b & B_a{}^d \\ B_b{}^c & G^{cd} \end{bmatrix},
\tag{22}
$$

where $G^{ab}$ is simply the inverse of $G_{ab}$. Since the whole background is defined by structure constants $f_{ab}{}^c$, it is convenient to work in the so-called generalised flux formulation of DFT [47]. For that, one defines generalised flux components $\mathcal{F}_{AB}{}^C$ simply as

$$[E_A, E_B]^M = \mathcal{F}_{AB}{}^C E_C{}^M, \tag{23}$$

where $E_A{}^M$ is the inverse generalised vielbein and the bracket is defined with respect to the generalized Lie derivative. Explicitly, one has $\mathcal{F}_{ABC} = 3E_{[A}{}^M E_B{}^N \partial_M E_{C]N}$. For a background given by a group manifold, one finds the only non-vanishing component $\mathcal{F}_{ab}{}^c = f_{ab}{}^c$. As expected, this is proportional to the structure constants of the algebra of the 1-forms $\sigma^a$. Given that the initial background is a solution to supergravity equations, such generalised flux and the undressed «flat» generalised metric $\mathcal{H}_{AB}$ are supposed to satisfy the equations of DFT in the flux formulation [47].

For the NATD procedure, one starts with the undressed fields packed into the "flat" generalised metric $\mathcal{H}_{AB}$ and first preforms a B-shift, that can be encoded as

$$\mathcal{H}'_{AB} = O_A{}^C(\tilde{y}) O_B{}^D(\tilde{y}) \mathcal{H}_{CD}, \quad O_A{}^B = \begin{bmatrix} \delta_a{}^b & 0 \\ \tilde{y}_e f_{ab}{}^e & \delta_c{}^d \end{bmatrix}. \tag{24}$$

Note that, since the dual coordinates carry the same indices $a, b$ as those labelling Lie algebra generators, the generalised metric $\mathcal{H}'_{AB}$ can be understood as the full generalised metric in curved indices. As we show below, one can think of $\mathcal{H}'_{AB}$ as such a generalised metric that linearly depends on dual coordinates and gives precisely the same generalised fluxes as the initial one. In this context, NATD is a transformation between two generalised metrics corresponding to the same generalised flux components.

Now, the crucial observation here is that $\mathcal{H}'_{AB}$ by construction generates precisely the same generalised flux $\mathcal{F}_{AB}{}^C$ as the initial background. This implies that $\mathcal{H}'_{AB}$ is a solution to the equations of double field theory. The last step is to turn dual coordinates $\tilde{y}_a$ into geometric coordinates by formal abelian T-dualities along all direction. This makes $\mathcal{H}'_{AB}$ a supergravity background, which has no dependence on dual coordinates and solves standard supergravity equations. It is worth mentioning, however, that after T-dualities, generalised flux components change, and one finds non-vanishing $\mathcal{F}_c{}^{ab}$ components, since T-duality along each direction replaces $_a \leftrightarrow {}^a$ [48]. The advantage of this language is that it can be directly generalised to an 11-dimensional background using the exceptional field theory construction.

It is important to mention the subtlety here related to the nature of the indices of the generalised metric $\mathcal{H}_{AB}$. Initially, these are in some sense doubled algebraic indices of the left-invariant 1-forms $\sigma^a$, while coordinates on the initial group manifold $y^m$ are labelled by "curved" indices $m, n, \ldots$. Looking at (21), one may say that formal 1-forms $d\tilde{y}_a$ are dual to the geometric 1-forms $\sigma^a$. It is not completely clear what is the strict mathematical meaning of a 1-form on the doubled space.

Before turning to an illustrative example, one observes that the last step where all directions of the group manifold get T-dualised is crucial for ending up with a supergravity solution. It is clear that one is always able to perform the necessary set of T-dualities to turn all $\tilde{y}_a$ into geometric coordinates. The picture however gets more complicated in the case of non-abelian U-dualities, and such a set may not exist. We discuss this important point in more detail in Section 4.

### 2.3. Bianchi II Example

As an explicit illustration of the above procedure, consider the standard example of Bianchi II cosmological space–time embedded into 10 dimensions. The metric can be chosen to be

$$\begin{aligned}
ds^2 = ds_6^2 &- a_1{}^2 a_2{}^2 a_3{}^2 (dx^0)^2 \\
&+ a_1{}^2 (\sigma^1)^2 + a_2{}^2 (\sigma^2)^2 + a_3{}^2 (\sigma^3)^2,
\end{aligned} \tag{25}$$

where the 1-forms $\sigma^a$ and the functions $a_a$ read

$$\sigma^1 = dy^1 - y^3 dy^2, \quad a_1{}^2 = \frac{p_1}{\cosh(p_2 x^0)}$$
$$\sigma^2 = dy^2, \qquad\qquad a_2{}^2 = \cosh(p_1 x^0) e^{p_2 x^0},$$
$$\sigma^3 = dy^3, \qquad\qquad a_3{}^2 = \cosh(p_1 x^0) e^{p_3 x^0}, \tag{26}$$

and the constants are constrained by $p_2 p_3 = p_1^2$. Such a defined metric solves supergravity equations when $p_a = 1$, which we assume from now on. Otherwise, one has to include non-trivial dilaton. Note that the 1-forms only depend on the coordinates $y^1, y^2, y^3$ on the group manifold generated by the Heisenberg–Weyl algebra:

$$d\sigma^a = f_{bc}{}^a \sigma^b \wedge \sigma^c, \quad f_{23}{}^1 = 1. \tag{27}$$

The undressed metric is then

$$||g_{ab}|| = \mathrm{diag}\left[-a_1{}^2 a_2{}^2 a_3{}^2, a_1{}^2, a_2{}^2, a_3{}^2, 1, \ldots, 1\right]. \tag{28}$$

Since the time direction $x^0$ is not dualised and the metric does not have mixed $g_{0a}$ components, it is enough to focus only on the block $1, 2, 3$ and consider $O(3, 3)$ double field theory. The corresponding generalised metric is simply given by

$$\mathcal{H}_{AB} = \begin{bmatrix} g_{ab} - B_{ae} g^{ef} B_{fb} & B_a{}^d \\ B_b{}^c & g^{cd} \end{bmatrix}, \tag{29}$$

where capital Latin indices from the beginning of the alphabet $A, B, \ldots$ represent doubled indices of undressed fields. The B-shift is performed by the matrix:

$$O^A{}_B = \begin{bmatrix} \delta^a{}_b & 0 \\ \Delta B_{cb} & \delta^d{}_c \end{bmatrix} \tag{30}$$

with $\Delta B_{ab} = \tilde{y}_c f_{ab}{}^c$, whose only non-vanishing components are

$$\Delta B_{23} = \tilde{y}_1. \tag{31}$$

Next, one is supposed to perform abelian T-dualities along all directions $\tilde{y}_a$. T-dualising along all three directions renders all $y^a$ non-geometric, as well as the corresponding forms, and one reproduces the well-known result for the dual background [25]:

$$ds'^2 = ds_6^2 - a_1{}^2 a_2{}^2 a_3{}^3 (dx^0)^2,$$
$$+ \frac{1}{a_1{}^2}(d\tilde{y}_1)^2 + \frac{a_3{}^2}{\Delta^2}(d\tilde{y}_2)^2 + \frac{a_2{}^2}{\Delta^2}(d\tilde{y}_3)^2$$
$$B' = -\frac{\tilde{y}_1}{\Delta^2} d\tilde{y}_2 \wedge d\tilde{y}_3,$$
$$\Delta^2 = a_2{}^2 a_3{}^2 + \tilde{y}_1^2. \tag{32}$$

The dilaton is recovered from the invariant dilaton:

$$e^{-2\phi}\sqrt{g} = e^{-2d} = e^{-2\phi'}\sqrt{g'}, \tag{33}$$

where $g = \det ||g_{ab}||$ is determinant of the undressed metric. Note that $\tilde{y}_a$ are now proper physical coordinates as these measure distances in the usual space–time. Indeed, before dualisation, the metric depends on $\tilde{y}_a$, while space–time shifts are $dy^a$, which follows from the pseudo-interval $\mathcal{H}_{AB} d\mathbb{X}^A d\mathbb{X}^B$. The dualisation is then understood as turning the coordinates $\tilde{y}_a$ into those, measuring distances in the physical space–time. The dependence

of fields on $\tilde{y}_a$ evidently does not change, only the meaning of the coordinate and the corresponding decomposition of the pseudo-interval. This understanding of dualities was suggested in [40,46] and further developed for non-geometric branes in [41,49,50].

*2.4. Partial NATD*

The example considered in the previous section suggests that in certain cases, one does not need to perform T-dualities along all directions in order to end up with a solution of supergravity equations. Indeed, consider a group generated by $T_I = (T_\alpha, T_a)$ with structure constants $f_{IJ}{}^K$ such that $f_{IJ}{}^\alpha = 0$. In this case, the DFT solution constructed by the B-shift as described above will not depend on $\tilde{y}_\alpha$. Hence, only dualisation along $T_a$ is required to turn $\tilde{y}_a$ into geometric coordinates. Starting with the Bianchi II cosmological solution and performing only T-duality along $\tilde{y}_1$, one ends up with the following background:

$$
\begin{aligned}
ds^2 &= ds_6{}^2 - a_1{}^2 a_2{}^2 a_3{}^2 (dx^0)^2 \\
&\quad + a_1{}^{-2}(d\tilde{y}_1)^2 + a_2{}^2(dy^2)^2 + a_3{}^2(dy^3)^2, \\
B &= \tilde{y}_1 dy^2 \wedge dy^3, \\
e^{-2\phi} &= a_1{}^2,
\end{aligned}
\tag{34}
$$

which satisfies the standard supergravity equations.

Let us look at how the Manin triple decomposition of the underlying Drinfeld algebra changes under the above transformation. We start with

$$
\begin{aligned}
[T_2, T_3] = T_1, \quad [T_2, \tilde{T}^1] = -\tilde{T}^3, \\
[T_3, \tilde{T}^1] = \tilde{T}^2.
\end{aligned}
\tag{35}
$$

Denoting $\mathfrak{g} = \mathrm{Span}(T_1, T_2, T_3)$ and $\tilde{\mathfrak{g}} = \mathrm{Span}(\tilde{T}^1, \tilde{T}^2, \tilde{T}^3)$ subalgebras spanned by the untilded and tilded generators, commutation relations can be written as

$$
[\mathfrak{g}, \mathfrak{g}] \subset \mathfrak{g}, \quad [\mathfrak{g}, \tilde{\mathfrak{g}}] \subset \mathfrak{g} \oplus \tilde{\mathfrak{g}}, \quad [\tilde{\mathfrak{g}}, \tilde{\mathfrak{g}}] \subset \tilde{\mathfrak{g}}.
\tag{36}
$$

The non-abelian T-duality transformation associated with the C-matrix that changes $T_1 \leftrightarrow \tilde{T}^1$ corresponds to making a different choice of the subset of generators $(\mathfrak{g}', \tilde{\mathfrak{g}}')$: $\mathfrak{g}' = \mathrm{Span}(\tilde{T}^1, T_2, T_3)$, $\tilde{\mathfrak{g}}' = \mathrm{Span}(T_1, \tilde{T}^2, \tilde{T}^3)$. In this case, we have

$$
[\mathfrak{g}', \mathfrak{g}'] \subset \tilde{\mathfrak{g}}', \quad [\mathfrak{g}', \tilde{\mathfrak{g}}'] = 0, \quad [\tilde{\mathfrak{g}}', \tilde{\mathfrak{g}}'] = 0,
\tag{37}
$$

which implies that $\mathfrak{g}'$ is not a geometric subalgebra and further (formal abelian) T-dualities are needed. Hence, although the operation generates solutions to supergravity equations, it is not described by the symmetries of a Drinfeld double. Indeed, in principle, one does not expect that any symmetry of string vacua must correspond to equivalence relations between Manin triples.

More generally, consider a Drinfeld double generated by $T_I = (T_a, T_\alpha)$ defined by

$$
\begin{aligned}
[T_I, T_J] = f_{IJ}{}^a T_a, \quad [T_I, \tilde{T}^a] = -f_{IK}{}^a \tilde{T}^K, \\
[\tilde{T}^I, \tilde{T}^J] = 0.
\end{aligned}
\tag{38}
$$

Since the dual structure constants $\tilde{f}_I{}^{JK} = 0$, the transformation $T_I \leftrightarrow \tilde{T}^I$ corresponds to NATD, and the group generated by $T_I$ is an isometry of the corresponding sigma-model. Following the above logic, let us now instead T-dualise only $T_a \leftrightarrow \tilde{T}^a$. This produces the following algebra:

$$[T'^a, T'^b] = 0, \qquad\qquad [T'^b, \tilde{T}'_a] = f_{ac}{}^b T'^c + f_{a\alpha}{}^b \tilde{T}^\alpha,$$
$$[T_\alpha, T'^a] = -f_{\alpha b}{}^a T'^b - f_{\alpha\beta}{}^a \tilde{T}^\beta, \quad [T_\alpha, \tilde{T}'_a] = -f_{a\alpha}{}^b \tilde{T}'_b$$
$$[T_\alpha, T_\beta] = f_{\alpha\beta}{}^a \tilde{T}'_a, \qquad\qquad [T'^a, \tilde{T}^\alpha] = 0, \tag{39}$$
$$[\tilde{T}'_a, \tilde{T}'_b] = f_{ab}{}^c \tilde{T}'_c,$$
$$[\tilde{T}^\alpha, \tilde{T}^\beta] = 0,$$

which is again not of the Drinfeld double type. Hence, one concludes that although partial non-abelian T-duality produces valid supergravity solutions, the corresponding transformation of isometry generators is not a symmetry of a Drinfeld double.

Notice that, since for the abelian case, commutation relations are trivial,

$$[\mathfrak{g}, \mathfrak{g}] = 0, \quad [\mathfrak{g}, \tilde{\mathfrak{g}}] = 0, \quad [\tilde{\mathfrak{g}}, \tilde{\mathfrak{g}}] = 0, \tag{40}$$

one is free to choose any subset of generators (given the projection constraint is satisfied) as the geometric subalgebra and no issues with partial dualisation appear. It is useful to look here at the Maurer–Cartan equations

$$d\sigma^a = f_{IJ}{}^a \sigma^I \wedge \sigma^J = f_{bc}{}^a \sigma^b \wedge \sigma^c + 2f_{b\alpha}{}^a \sigma^b \wedge \sigma^\alpha + f_{\alpha\beta}{}^a \sigma^\alpha \wedge \sigma^\beta,$$
$$d\sigma^\alpha = 0. \tag{41}$$

As before, this means that dual coordinates $\tilde{y}_\alpha$ do not enter the B-shift and, hence, no (formal abelian) T-duality in this direction is needed to arrive at a solution. Looking at the Killing vector algebra:

$$[k_I, k_J] = f_{IJ}{}^K k_K, \tag{42}$$

one notices that $\mathfrak{a} = \{k_a\}$ is a subalgebra of isometries. However, since the forms $\sigma^a$ do not trivially disentangle from the rest $\sigma^\alpha$, the NATD procedure cannot be applied. Moreover, since in general, the set of MC forms as defined above does not correspond to that of a coset space, the corresponding NATD procedure as described in [7] also does not apply.

Interestingly, partial non-abelian T-duality is a symmetry of the two-dimensional sigma-model. Indeed, consider as before a two-dimensional sigma-model on a manifold of the form $M \times G$ where $G$ is a group manifold corresponding to an algebra $\mathfrak{g}$ with generators $T_I = (T_a, T_\alpha)$. Suppose further that $f_{IJ}{}^\alpha = 0$, which implies that the Maurer–Cartan forms $\sigma^\alpha = dy^\alpha$. Following the standard procedure, we replace the rest of the forms $\sigma^a \to A^a$ and add Lagrange terms $\tilde{y}_a F^a$:

$$S = \int_{\Sigma_2} dx^\mu (G_{\mu\nu} * + B_{\mu\nu}) \wedge dx^\nu + 2dy^\alpha (G_{\alpha\mu} + B_{\alpha\nu}) \wedge dx^\mu + dy^\alpha (G_{\alpha\beta} * + B_{\alpha\beta})$$
$$+ 2A^a (G_{a\mu} * + B_{a\mu}) \wedge dx^\mu + 2A^a (G_{a\alpha} * + B_{a\alpha}) \wedge dy^\alpha \tag{43}$$
$$+ A^a (G_{ab} * + B_{ab}) \wedge A_b + \tilde{y}_a F^a,$$

where

$$F^a = dA^a + f_{bc}{}^a A^b \wedge A^c + 2f_{\alpha b}{}^a A^\alpha \wedge dy^b + f_{\alpha\beta}{}^a dy^\alpha \wedge dy^\beta. \tag{44}$$

Integrating out the Lagrange multiplier $\tilde{y}_a$, one obtains the equation of motion $F^a = 0$, which implies $A^a = \sigma^a$. Alternatively, one may integrate out the fields $A^a$, for which it is convenient to combine $x^m = (x^\mu, y^\alpha)$ and to rewrite the action as follows:

$$S = \int_{\Sigma_2} dx^m (G_{mn} * + \hat{B}_{mn}) \wedge dx^n + 2A^a (G_{am} * + \hat{B}_{am}) \wedge dx^m$$
$$+ A^a (G_{ab} * + B_{ab}) \wedge A_b + \tilde{y}_a (dA^a + f_{bc}{}^a A^b \wedge A^c), \tag{45}$$

where we define

$$
\hat{B}_{\mu\nu} = B_{\mu\nu}, \quad \hat{B}_{\alpha\mu} = B_{\alpha\mu}, \qquad \hat{B}_{\alpha\beta} = B_{\alpha\beta} + \tilde{y}_a f_{\alpha\beta}{}^a,
$$
$$
\hat{B}_{a\mu} = B_{a\mu}, \quad \hat{B}_{a\alpha} = B_{a\alpha} + f_{a\alpha}{}^b \tilde{y}_b
$$

(46)

Solving equations for $A^a$, e.g., as in [29], and substituting back into the action, one obtains an equivalent sigma-model on a different background:

$$
G'_{mn} = G_{mn} - (G - \hat{B})_{a(m} N^{ab} (G + \hat{B})_{n)b}
$$
$$
G'_{ma} = \frac{1}{2}(G - \hat{B})_{mb} N^{ba} - \frac{1}{2} N^{ab}(G - \hat{B})_{bm}
$$
$$
G'_{ab} = N^{(ab)}
$$
$$
B'_{mn} = \hat{B}_{mn} + (G - \hat{B})_{a[m} N^{ab} (G + \hat{B})_{n]b}
$$
$$
B'_{ma} = -\frac{1}{2}(G - \hat{B})_{mb} N^{ba} - \frac{1}{2} N^{ab}(G - \hat{B})_{bm}
$$
$$
B'_{ab} = -N^{[ab]}
$$

(47)

with $N_{ab} = G_{ab} - B_{ab} + \tilde{y}_c f_{ab}{}^c$. These are simply Buscher rules for T-dualities along $\tilde{y}_a$. The dilaton is then transformed such that the combination $d = \phi - \frac{1}{4}\log\det G$ remains invariant.

Hence, one concludes that for the particular case $f_{IJ}{}^\alpha = 0$, i.e., when $\sigma^\alpha = dy^\alpha$, to obtain an equivalent $\sigma$-model, it is enough to T-dualise only the remaining directions $\tilde{y}_a$. This is completely natural from the point of view of DFT as the background depends on $\tilde{y}_a$ and does not depend on $\tilde{y}_\alpha$. However, although such duality generates solutions to equations of the standard supergravity, the transformation of the corresponding Manin triple does not give a Manin triple. Hence, the transformation is not a symmetry of the Drinfeld double, and "partial" T-duality, although generating proper backgrounds, does not make as much sense as the usual (N)ATD, which is based on the symmetries of the underlying Drinfeld algebra. The same observation can be made for NAUD, on which we comment later.

### 3. Non-Abelian U-Duality in SL(5) ExFT

Let us now try to generalise the above algorithm of NATD to the case of exceptional field theory. As the very first example, one may take the SL(5) exceptional field theory, that is a $7 + 10$-dimensional field theory, whose local coordinate transformations include U-dualities of $D = 7$ maximal supergravity [51,52] (for a review on exceptional field theories, see [53–55]). Space–time is split into 7 external directions labelled by coordinates $x^\mu$, 4 internal coordinates $y^m$, and 6 dual coordinates $\tilde{y}_{mn} = -\tilde{y}_{nm}$ corresponding to winding modes of the M2-brane. The latter form the 10-dimensional extended space parametrised by $\mathbb{X}^{MN} = -\mathbb{X}^{NM}$, on which the generalised Lie derivative is defined. Capital Latin indices label the irrep **5** of SL(5). The closure of the algebra of generalised Lie derivatives imposes the section condition on all fields and their combinations, which schematically can be written as

$$
\epsilon^{MNKLP} \partial_{MN} \bullet \partial_{KL} \bullet = 0.
$$

(48)

The field content of the theory can be written in irreps of the duality group SL(5) as follows:

$$
g_{\mu\nu}, \quad A_\mu{}^{[MN]}, \quad m_{(MN)}, \quad B_{\mu\nu M},
$$

(49)

where the generalised metric $m_{MN}$ parametrises the coset space SL(5)/SO(5). Explicit parametrisation in terms of supergravity fields depends on the choice of the frame, which is dictated by the choice of the solution to the section constraint. In addition to the straight-forward minimal choice $\partial_{MN} = 0$ giving $D = 7$ ungauged maximal supergravity, one finds two distinct maximal solutions of the section constraint. These correspond to breaking of the set $\mathbb{X}^{MN}$ labelling the **10** of SL(5) with respect to subgroups GL(4) and GL(3) × SL(2). The

former turns SL(5) ExFT into 11d supergravity, while the latter gives Type IIB supergravity in the S-duality covariant formulation.

For the purpose of this paper, we are interested in relations between fields in 11D and IIB frames recovered from explicit parametrisations of the generalised metric $m_{MN}$ and the relation of the external metric $g_{\mu\nu}$ to the $7 \times 7$ block of the full 11/10-dimensional metric. One starts with the 11-dimensional metric written in the $7 + 4$-split:

$$
\begin{aligned}
ds_{11}^2 &= \hat{g}_{\mu\nu} dx^\mu dx^\nu + \hat{h}_{mn} dy^m dy^n \\
&= \hat{g}_{\mu\nu} dx^\mu dx^\nu + \hat{h}_{ab} \sigma^a \sigma^b.
\end{aligned}
\tag{50}
$$

Then, one has for the ExFT fields $g_{\mu\nu}$ and $m_{AB}$ [56,57]:

$$
\begin{aligned}
g_{\mu\nu} &= \hat{h}^{\frac{1}{5}} \hat{g}_{\mu\nu}, \\
m_{AB} &= \hat{h}^{\frac{1}{10}} \begin{bmatrix} \hat{h}^{-\frac{1}{2}} \hat{h}_{ab} & V_a \\ V_b & \hat{h}^{\frac{1}{2}} (1 + V^2) \end{bmatrix},
\end{aligned}
\tag{51}
$$

where $\hat{h} = \det ||\hat{h}_{ab}||$ and the vector $V^a$ encodes internal components of the 3-form field $V^a = \hat{h}^{-\frac{1}{2}} \epsilon^{abcd} C_{bcd}$. Note that $\det m_{AB} = 1$ and is parametrised by undressed fields.

To recover fields of Type IIB supergravity one switches to the parametrisation corresponding to the $\mathrm{GL}(3) \times \mathrm{SL}(2)$ solution of the section constraint, keeping the ExFT fields the same. For that, one has

$$
\begin{aligned}
g_{\mu\nu} &= e^{-\frac{4}{5}d} \tilde{g}_{\mu\nu}, \\
m_{AB} &= e^{-\frac{2}{5}d} \begin{bmatrix} \tilde{h}^{\frac{1}{2}} \tilde{h}^{ab} + e^{-2d} \mathcal{M}^{ij} V_i^a V_j^b & V_i^a \\ V_j^b & e^{2d} \mathcal{M}_{ij} \end{bmatrix}, \\
e^{-2d} &= e^{-2\phi} \tilde{h}^{\frac{1}{2}}.
\end{aligned}
\tag{52}
$$

Here, $d$ is the invariant dilaton of double field theory, $\tilde{h}_{ab}$ is the 3-dimensional block of the full 10-dimensional metric, and the matrix $\mathcal{M}_{ij}$ encodes the degrees of freedom of the axion–dilaton:

$$
||\mathcal{M}_{ij}|| = \begin{bmatrix} 1 & C_0 \\ C_0 & e^{-2\phi} + C_0^2 \end{bmatrix}.
\tag{53}
$$

The pair of vectors $V_i^a$ encodes the internal parts of the NS-NS Kalb–Ramond 2-form $B_{ab}$ and R-R field $C_{ab}$ as

$$
V_i^a = \tilde{h}^{-1/2} \epsilon^{abc} \begin{bmatrix} C_{bc} \\ B_{bc} \end{bmatrix},
\tag{54}
$$

where $\epsilon^{abc}$ is the Levi-Civita symbol $\epsilon^{123} = 1$. It is important to notice that the parametrisation used here differs from that of [33] as we are working in the string frame. The parametrisation in the Einstein frame of [33] provides the formulation of IIB supergravity explicitly covariant under the SL(2) duality symmetry, which is reflected in the fact that all dependence on the dilaton is hidden inside the SL(2)/SO(2) matrix and the metric is inert under the S-duality. In contrast, the parametrisation given above provides fields T-dual to the IIA fields, which can be obtained from the standard 11D parametrisation. For the purposes of this paper, the latter is more convenient.

Now, following the analogy between DFT and ExFT extended spaces, were propose the following non-abelian U-duality scheme for 11D backgrounds:

1. Undress the metric and the C-field $g_{mn} = \sigma_m^{\ a} \sigma_n^{\ b} g_{ab}$, $C_{mnk} = \sigma_m^{\ a} \sigma_n^{\ b} \sigma_k^{\ c} C_{abc}$, and compose generalised metric $\mathcal{H}_{AB}$ from the undressed fields.
2. Construct a background with a C-field defined by $C_{abc} + \Delta C_{abc}$ with the shift given by $\Delta C_{abc} = -3\tilde{y}_{d[a} f_{bc]}^{\ \ d}$, where $\tilde{y}_{ab}$ are the would-be dual coordinates. The metric is then simply the flat metric $g_{ab}$.

3.　Perform a U-duality transformation that turns $\tilde{y}_{ab}$ into geometric coordinates and $\sigma^a$ into dual 1-forms. Equivalently, embed gl(4) in a different way.

Step 3 above needs further clarification. Notice first that the background constructed at Step 2 is such that it solves the equations of ExFT whenever the initial background is a supergravity solution. The reason is simply that these two backgrounds have identical generalised fluxes. Due to the explicit dependence on dual coordinates in the C-field, the constructed background cannot solve the equations of supergravity and has to be U-dualised into a proper supergravity background at Step 3. Note that, in the case of the degenerate Cartan–Killing form, less coordinates appear in the C-shift. For example, for $f_{ab}{}^c = 0$, the background does not depend on dual coordinates at all, and formally, no dualisation is needed. Indeed, this is simply a flat background with constant C-field $(g_{ab}, C_{abc})$, which apparently satisfies the equations. The initial background would then be a flat torus. The important observation here, to be elaborated on below, is that such a dualisation of all coordinates is not always possible. For example, for the SL(5) case, which we consider in more detail below, one is not able to U-dualise four non-geometric coordinates into four geometric coordinates by an external automorphism. However, this is possible for a subset of three, requiring the remaining direction to be a separate U(1) isometry. In this case, the procedure simply reproduces non-abelian T-duality and relates the IIA and IIB backgrounds.

As in the case of partial NATD discussed in Section 2.4, structure constants $f_{ab}{}^c$ may be such that, upon the shift, the C-field depends on less than four dual coordinates, while the background does not have a separate U(1) direction. In this case, U-dualisation of less than four coordinates is needed to end up with a solution to equations of the standard 11-dimensional supergravity. However, for the same reason, this will not be a symmetry of EDA, since the multiplication $\mathfrak{g} \circ \mathfrak{g}$ will have generators from the dual $\tilde{\mathfrak{g}}$ on the RHS. Although these are interesting to discuss as symmetries of the corresponding sigma-model, we will not consider such partial NAUD's in what follows.

As in the case of NATD transformations represented as a B-shift plus T-dualities, the above procedure guarantees always giving a solution to the equations of 11D supergravity, however with an additional restriction to unimodular groups, i.e., $f_{ab}{}^b = 0$. The origin of the latter condition will become clear momentarily. This procedure is in the same relation to Nambu–Lie U-duality described in [30,58] as NATD is to Poisson–Lie T-duality. More concretely, for a Poisson–Lie T-duality transformation, one starts with a given realisation of a Drinfeld double $\mathcal{D}$ and searches for such an O$(d, d)$ matrix $C_A{}^B$ that transforms generators $T_A$ of $\mathcal{D}$ into a new set $T'_A$ also defining a Drinfeld double. For the particular choice of $C_A{}^B$ corresponding to the B-shift+T-dualities' transformation of the initial background, the procedure is commonly referred to as non-abelian T-duality, as it is performed over a group manifold background. Similarly, Nambu–Lie U-duality is concerned with searches of such $C_A{}^B \in \mathrm{E}_{d(d)}$ preserving the structure of exceptional Drinfeld algebra. As noted in [33,58], this step is very complicated. Instead, the procedure suggested above provides explicit field transformation rules, similar to Buscher rules. Explicit examples of non-abelian U-dual backgrounds based on this procedure are provided in the separate work [34].

The proof that one always ends up with a solution is a straightforward repetition of that for double field theory. Starting with a solution to 11D equations, one undresses all fields and composes a "flat" generalised metric $m_{AB}$ of exceptional field theory. The corresponding generalised vielbein $E_M{}^A$ contains only components of the left-invariant 1-forms $\sigma_m{}^a$, and hence, the only non-vanishing generalised flux components are those, proportional to the structure constants $f_{ab}{}^c$. Using the usual expressions for fluxes of the SL(5) ExFT as in [59,60], one finds for the components of **10**, **15**, and $\overline{\mathbf{40}}$:

$$S_{a5} = 2f_{ba}{}^b, \quad \theta_{a5} = \frac{1}{2}f_{ba}{}^b, \quad \tilde{T}_{ab5}{}^c = -f_{ab}{}^c - \frac{2}{3}\delta_{[a}{}^c f_{b]d}{}^d. \tag{55}$$

As before, C-shift turning $m_{AB}$ to $m'_{AB}$ can be understood as a generalised vielbein with the following generalised fluxes:

$$S'_{a5} = 4f_{ba}{}^b, \quad \theta'_{a5} = -f_{ba}{}^b, \quad \tilde{T}'_{ab5}{}^c = -f_{ab}{}^c - \frac{2}{3}\delta_{[a}{}^c f_{b]d}{}^d. \tag{56}$$

Hence, by construction, generalised fluxes of such a constructed background with a flat metric and the C-field linearly depending on dual coordinates $y_{ab}$ are *the same* as that of the initial background, given $f_{ab}{}^b = 0$. Up to this condition, in terms of generalised fluxes and the undressed generalised metric $m_{AB}$, nothing has been changed, and the background $m'_{AB}$ solves the equations of ExFT. Arguments for the case, where undressed fields depend on external coordinates $m_{AB} = m_{AB}(x)$, go along the same lines, with however more involved equations in the flux formulation of ExFT. Finally, U-dualising all winding coordinates entering the linear dependence, one ends up with a solution of 11D equations. This step appears to be the most tricky.

Although here, we restrict ourselves to the case of SL(5) ExFT for simplicity, the first two steps of the procedure have a straightforward generalisation to a higher U-duality group simply by including more winding coordinates. In contrast, the last step appears to be much more restricted for the SL(5) theory than for theories with more winding directions. As we show below, at least for group manifolds, full dualisation of all four coordinates is possible only when at least one of the coordinates is an abelian isometry. We conclude that the described procedure for the SL(5) ExFT is always an uplift of an NATD transformation. Similar observations based on the construction of exceptional Drinfeld algebras for the SL(5) theory were made in [33]. Schematically, this is illustrated in Figure 2.

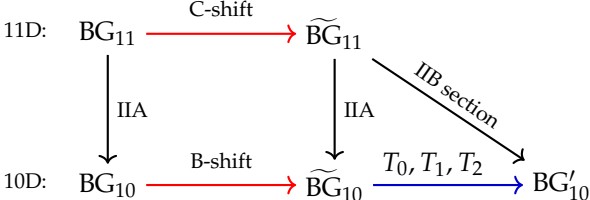

**Figure 2.** Relationship between backgrounds with spectator fields upon the non-abelian U-duality procedure. Here, taking an IIB section represents an uplift of three T-dualities with further reduction to 10 dimensions. In this case, the bottom line represents the usual non-abelian T-duality.

## 4. Algebraic Perspective

### 4.1. T-Duality

Non-abelian T-duality as a particular case of Poisson–Lie T-duality is based on the notion of Drinfeld double Lie algebra, which is basically a Manin triple $(\mathfrak{g}, \tilde{\mathfrak{g}}, \eta)$, with $\mathfrak{g}$ and $\tilde{\mathfrak{g}}$ being Lie algebras with bases $\{T_a\} = \text{bas}\,\mathfrak{g}$ and $\{\tilde{T}^a\} = \text{bas}\,\tilde{\mathfrak{g}}$ and $\eta$ a non-degenerate invariant symmetric bilinear form on $\mathfrak{g} \oplus \tilde{\mathfrak{g}}$. Defining $\{T_A\} = \{T_a, \tilde{T}^a\} = \text{bas}\,\mathfrak{g} \oplus \tilde{\mathfrak{g}}$, the commutation relations read

$$[T_A, T_B] = \mathcal{F}_{AB}{}^C T_C, \tag{57}$$

where in general, the only non-vanishing structure constants are $\mathcal{F}_{ab}{}^c =: f_{ab}{}^c$ and $\mathcal{F}^{ab}{}_c =: \tilde{f}^{ab}{}_c$. For NATD, either $\mathfrak{g}$ or $\tilde{\mathfrak{g}}$ should be Abelian. Non-vanishing components of the *ad*-invariant symmetric form $\eta(T_A, T_B) = \eta_{AB}$ are then $\eta^a{}_b = \delta^a{}_b = \eta_b{}^a$.

Geometrically, such a defined Drinfeld double can be realised by choosing a maximally isotropic subalgebra, say $\mathfrak{g}$, to be a "physical" subalgebra. Group element $g = \exp x^a T_a$ of the corresponding Lie group $G$ defined by generators of the physical subalgebra will define left-invariant 1-forms $\sigma = g^{-1}dg$ on the group manifold. In this setup, a non-abelian T-duality corresponds to transfer the role of the physical subalgebra to the dual algebra $\tilde{\mathfrak{g}}$ and constructing space–time 1-forms from group element $\tilde{g} = \exp x_a \tilde{T}^a$. Note that, here, $x_a$ is a physical coordinate and no further T-duality is required. More generally, a Poisson–Lie T-duality is a constant transformation of the generators $T_A$ preserving the bilinear form $\eta$,

i.e., an O$(d,d)$ rotation $T'_A = C_A{}^B T_B$, under which a given Drinfeld double is invariant. A split of the Drinfeld double into sets of physical and dual coordinates breaks O$(d,d)$ into the physical GL$(d)$, upon which the vector representation decomposes as

$$\mathbf{2d} \longrightarrow \mathbf{d} \oplus \bar{\mathbf{d}}. \tag{58}$$

The $\mathbf{d}$ will be chosen to correspond to the physical subalgebra, while the alternative choice corresponds to taking $\bar{\mathbf{d}}$. The corresponding physical algebra $\mathfrak{gl}(d)$ will be embedded into $\mathfrak{o}(d,d)$ differently, and switching between these two is represented by the external automorphism of $\mathfrak{o}(d,d)$ corresponding to switching the two spinorial roots.

More transparently, this is seen when looking at generalised vielbeins $E_A{}^M$ with the inverse defining the generalised metric $\mathcal{H}_{MN} = E_M{}^A E_N{}^B \mathcal{H}_{AB}$. These can be explicitly constructed in the component form in terms of the left-invariant 1-forms and a $B$- or $\beta$-field. For us, it important is that the generalised vielbein $E_A{}^M$ realises the same double Drinfeld algebra with respect to to the generalised Lie derivative (see [35] for more details):

$$[E_A, E_B] = \mathcal{F}_{AB}{}^C E_C. \tag{59}$$

Hence, a transformation $C_A{}^B$ can be understood as acting on the algebraic indices of the generalised vielbein. As was explicitly shown at the level of the sigma-model in [29], an NATD transformation is precisely a B-shift and a series of O$(d,d)$ reflections along all $d$ directions, performed on the undressed generalised vielbein $E_M{}^A$. From the DFT point of view, the latter are necessary to turn all $\tilde{x}_a$ into geometric coordinates, while in the Drinfeld double language, this replaces all generators $T_a$ by $\tilde{T}^a$. The set of $d$ T-dualities interchanging $\mathbf{d}$ and $\bar{\mathbf{d}}$ (normal and winding coordinates) can be equivalently understood as choosing a different embedding of the maximal GL$(d)$ subgroup, such that $\bar{\mathbf{d}}$ becomes its fundamental and $\mathbf{d}$ its co-fundamental representations. This indeed corresponds to changing the deleted spinorial root of the Dynkin diagram of O$(d,d)$ and, hence, to the external automorphism.

One notices that, according to the B-shift+T-dualities procedure, one has to replace all winding coordinates by their geometric partners, which can be done in a unique way for O$(d,d)$ theory (for group manifolds that are not a product of Lie groups). This seems to be in tension with the Poisson–Lie T-plurality picture, where a given Drinfeld double can be decomposed into a set of more than two Manin triples [61]. Backgrounds corresponding to such Manin triples generate the same Drinfeld double and hence are indistinguishable from the point of view of the two-dimensional sigma-model. Examples of such backgrounds can be found in [26]. In the O$(d,d)$ language, Poisson–Lie T-plurality corresponds to performing a rotation by an O$(d,d)$ matrix $C_A{}^B$, preserving the Drinfeld double, which in particular can be a set of $d$ reflections [35]. This latter case is precisely the transformation, which turns all winding coordinates into geometric ones. Hence, in all other cases, one would expect backgrounds that do not solve equations of normal supergravity due to the remaining dependence on the winding coordinates. Indeed, as shown on explicit examples in [26], such a procedure in particular gives solutions of generalised supergravity equations.

It is important to notice here that while preserving the section constraint, a general O$(d,d)$ transformation does not necessarily keep a background in the set of solutions to supergravity equations. The well-known example is the linear dilaton background of generalised supergravity. One starts with a background for which only the dilaton depends on a given coordinate, say $x^9$, while all other fields are isometric along it. Normally, T-duality is forbidden along non-isometric coordinates; however, DFT allows performing an O$(d,d)$ rotation turning $x^9$ into non-geometric $\tilde{x}_9$, and the dilaton becomes

$$\phi = f(x^0, \ldots, x^8) + c\tilde{x}_9. \tag{60}$$

Apparently, the section condition is preserved, as nothing depends on $x^9$, and the background still solves the DFT equations. However, the equations of supergravity are violated due to the dependence on the dual coordinate. Instead, since the dilaton always

enters the equations via its derivatives, the background can be shown to satisfy a deformed version of supergravity [42]. Another example of a background depending on dual coordinates in a section condition preserving way is given by localised exotic branes, considered in [41]. These are obtained by global $O(d,d)$ reflections and are solutions to DFT equations, while violating the supergravity equations.

For DFT backgrounds represented by group manifolds, equivalent $\mathfrak{gl}(d)$ embeddings into $\mathfrak{o}(d,d)$ can be obtained from a given one by $O(d,d)$ rotations and by the external automorphism of the algebra. Only the latter turns the fundamental of a given embedding of $\mathfrak{gl}(d)$ into the antifundamental of the dual embedding. Crucial here is that no weight belongs to both these representations, which is apparent for the $\mathfrak{o}(d,d)$ algebra, but is not always true for symmetry algebras of exceptional field theories.

To conclude, one starts with an irrep $\mathcal{R}_1$ of the abelian T(U)-duality group in which extended coordinates transform. Upon an embedding of the geometric $GL(d)$ subgroup, this decomposes into $\mathcal{R}_1 \to \mathbf{d} \oplus \dots$, where $\mathbf{d}$ corresponds to geometric coordinates and ellipses denote irreps under which winding coordinates transform. Now, one considers a different embedding of the geometric $GL(d)$ such that $\mathcal{R}_1 \to \mathbf{d}' \oplus \dots$, where $\mathbf{d}'$ is the fundamental of $GL(d)$, none of whose weights inside $\mathcal{R}_1$ coincide with that of $\mathbf{d}$. Let us provide more details for U-duality groups SL(5), where this cannot be done, and SO(5,5), which can be shown to allow 11-dimensional NAUD.

*4.2. U-Duality and Exceptional Drinfeld Algebras*

We start with the set of simple roots of the Lie algebra $\mathfrak{sl}(5)$ in the canonical $\omega$-basis of fundamental weights:

$$
\begin{aligned}
\alpha_{12} &= (2,-1,0,0), \\
\alpha_{23} &= (-1,2,-1,0), \\
\alpha_{34} &= (0,-1,2,-1), \\
\alpha_{45} &= (0,0,-1,2),
\end{aligned}
\tag{61}
$$

where labelling of the roots will become clear momentarily. The remaining positive roots are

$$
\begin{aligned}
\alpha_{13} &= \alpha_{12} + \alpha_{23}, & \alpha_{14} &= \alpha_{12} + \alpha_{23} + \alpha_{34}, \\
\alpha_{24} &= \alpha_{23} + \alpha_{34}, & \alpha_{25} &= \alpha_{23} + \alpha_{34} + \alpha_{45}, \\
\alpha_{35} &= \alpha_{34} + \alpha_{45}, & \alpha_{15} &= \alpha_{12} + \alpha_{23} + \alpha_{34} + \alpha_{45}.
\end{aligned}
\tag{62}
$$

In addition, one has the same number of negative roots and four Cartan generators. The weight diagram of the fundamental representation **5** of $\mathfrak{sl}(5)$ is depicted in Figure 3, where $\mu_1, \dots, \mu_5$ denote basis vectors. Notations for simple roots of the algebra are chosen in such a way that, say, the root $\alpha_{12}$ sends the weight vector $\mu_1$ to $\mu_2$ or, equivalently, the exponent $\exp(\omega \alpha_{12})$ acts by SL(2) rotations on the plane $(\mu_1, \mu_2)$.

From the Dynkin diagram of $\mathfrak{sl}(5)$ in Figure 4, one finds two embeddings of the subalgebra $\mathfrak{gl}(4)$, corresponding to deleting the root $\alpha_{12}$ or the root $\alpha_{45}$. In matrix representation, this corresponds to embedding a $4 \times 4$ matrix as an upper left or lower right block. As is shown in Figure 3, depending on the chosen deletion of a root, one ends up with different decompositions of the fundamental $\mathbf{5} \to \mathbf{4} \oplus \mathbf{1}$. It is important to note that the weights $\mu_2, \mu_3, \mu_4$ belong to a **4** for both of the decompositions, while one of $\mu_1$ and $\mu_5$ becomes a singlet.

Following the analogy with NATD, one is interested in embeddings of the physical $\mathfrak{gl}(4)$ subalgebra related by the external automorphism. In particular, for the SL(5) theory, we are interested in decomposing the **10** of $\mathfrak{sl}(5)$ upon two embeddings of $\mathfrak{gl}(4)$, which are shown in Figure 5. Consider first the decomposition corresponding to deleting the root $\alpha_{45}$ (cutting blue arrows). In this case, weight vectors $\mathbb{X}^{5a}$ with $a = 1, \dots, 4$ belong to the **4** of $\mathfrak{gl}(4)$, while the rest $\mathbb{X}^{ab}$ belong to the **6**. In the ExFT language, the former get identified with geometric coordinates, while the latter represent winding modes.

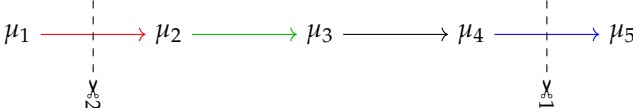

**Figure 3.** Weight diagram of the fundamental **5** of $\mathfrak{sl}(5)$ with the highest weights represented by $\mu_1$. The action of different roots is denoted by different colours, and the direction of arrows shows the lowering of the weight. Depending on the chosen deletion of a simple root, one obtains two different decompositions $\mathbf{5} \to \mathbf{4_0} + \mathbf{1_{-4}}$ under $\mathfrak{sl}(5) \hookleftarrow \mathfrak{gl}(4)$.

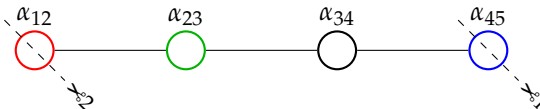

**Figure 4.** Dynkin diagram of $\mathfrak{sl}(5)$ with simple roots coloured differently for further convenience. Depending on the two possible ways to delete one root keeping three connected, depicted by $\rtimes 1$ and $\rtimes 2$, one obtains two embeddings of the $\mathfrak{gl}(4)$ subalgebra related by the external automorphism.

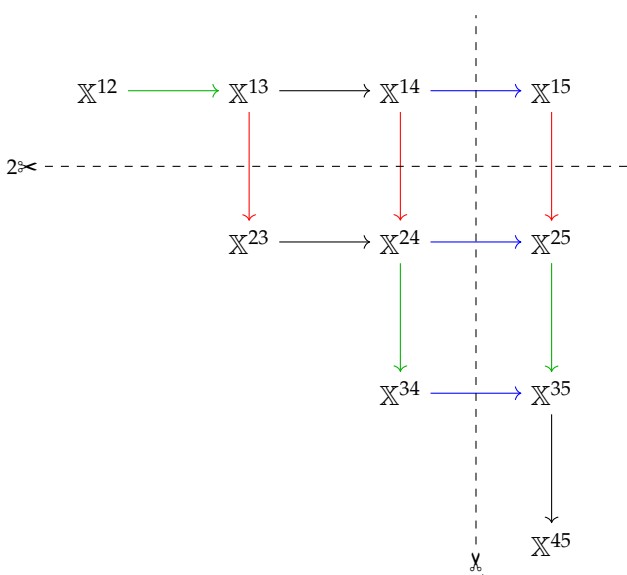

**Figure 5.** Weight diagram of the **10** of $\mathfrak{sl}(5)$ with two possible embeddings of the $\mathfrak{gl}(4)$ subalgebra.

Now, according to the procedure of NAUD described above, one needs to find such a different embedding of $\mathfrak{gl}(4)$ that all weights contributed to the irrep governing geometric coordinates of the first embedding belong to that governing winding modes. Explicitly, all weights from the old **4** must belong to the new **6**, which is impossible, according to Figure 5.

Indeed, suppose one starts with four left-invariant 1-forms $\sigma^a$ that depend on four coordinates on the (unimodular) group manifold $x^1, x^2, x^3, x^4$. Next, one constructs a background with a flat metric and C-field given by $C_{abc} = -3\tilde{y}_{d[a}f_{bc]}{}^d$ with $\tilde{y}_{ab} = 1/2\epsilon_{abcd}\mathbb{X}^{cd}$ being coordinates along winding directions. This has been shown to solve the equations of ExFT; however, to end up with an ordinary supergravity solution, one has to turn all $\tilde{y}_{ab}$ entering the linear dependence into geometric coordinates. From Figure 5, one observes that the external automorphism exchanging the roots $\alpha_{12}$ and $\alpha_{45}$ relates two different $\mathfrak{gl}(4)$ embeddings such that the weights labelled $(\mathbb{X}^{15}, \mathbb{X}^{25}, \mathbb{X}^{35}, \mathbb{X}^{45})$ form the **4** of the first $\mathfrak{gl}(4)$, while $(\mathbb{X}^{12}, \mathbb{X}^{13}, \mathbb{X}^{14}, \mathbb{X}^{15})$ form the **4** of the second $\mathfrak{gl}(4)$. Hence, to end up with the dependence of $C_{abc}$ on four geometric coordinates after turning from one $\mathfrak{gl}(4)$ to the other, one has to start with the dependence on the coordinates $(\mathbb{X}^{12}, \mathbb{X}^{13}, \mathbb{X}^{14}, \mathbb{X}^{15})$, as is clear from the weight diagram. However, $\mathbb{X}^{15}$ is defined as a geometric coordinate with respect to the first embedding of $\mathfrak{gl}(4)$ and, hence, must not enter the linear dependence. One is thus left

with dependence on only three coordinates $(\mathbb{X}^{12}, \mathbb{X}^{13}, \mathbb{X}^{14})$ in the C-field, which correspond to the duality M-IIB. Simple adding a term proportional to $\mathbb{X}^{5a}$ into the linear ansatz for the C-field will not work as there is no way to combine the irreps **4** and $\bar{\mathbf{4}} \oplus \overline{\mathbf{15}}$ into the **4**.

Put differently, one starts with a group manifold background defined by Maurer–Cartan forms $(\sigma^{15}, \sigma^{25}, \sigma^{35}, \sigma^{45})$ in ExFT notation. Upon the external automorphism, their duals are $(\sigma^{15}, d\mathbb{X}^{12}, d\mathbb{X}^{12}, d\mathbb{X}^{14})$, i.e., the $\sigma^{12}$ transforms into itself. On the other hand, the forms $\sigma^{5a}$ depend on coordinates $\mathbb{X}^{5a}$, which we call geometric. It is quite clear that after the duality, the coordinates $(\mathbb{X}^{25}, \mathbb{X}^{35}, \mathbb{X}^{45})$ become geometric, and according to the Maurer–Cartan equation:

$$d\sigma^{5a} = f_{bc}\sigma^{5b} \wedge \sigma^{5c} \tag{63}$$

the only way to avoid their appearance in $\sigma^{15}$ is to set $d\sigma^{15} = 0$. In other words, the form $\sigma^{15}$ should represent a separate U(1) or $\mathbb{R}$ isometry turning the transformation into NATD. Hence, one concludes that the described procedure applied to a 4-dimensional group manifold always provides a solution of 10-dimensional supergravity equations, which is consistent with the observations made in [33]. Another option would be to generalise the notion of T-plurality to the case of non-abelian U-duality. From the DFT point of view, T-plurality generates backgrounds with dependence on dual coordinates, which in particular cases solve generalised supergravity equations. However, no generalised supergravity extension to 11 dimensions is known, and moreover, this is widely accepted to not exist.

Consider now a more fruitful case of five dimensions and U-duality algebra $\mathfrak{so}(5,5)$. Its Dynkin diagrams with two possible deletions of simple roots giving $\mathfrak{gl}(5)$ is depicted in Figure 6. This has three simple roots generating the vector representation, antisymmetric tensor of second and third rangerepresentations, and two spinorial representations.

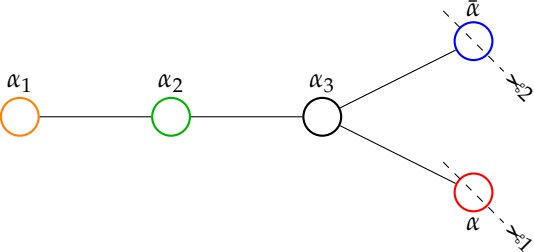

**Figure 6.** Dynkin diagram of $\mathfrak{so}(5,5)$ with simple roots coloured differently for further convenience. Depending on the two possible ways to delete one root keeping three connected, depicted by ⋊1 and ⋊2, one obtains two embeddings of the $\mathfrak{gl}(5)$ subalgebra.

Embeddings of $\mathfrak{gl}(5)$ are recovered by deleting one of the two spinorial roots: $\alpha$ or $\bar{\alpha}$. Here, we focus on the **16** of $\mathfrak{so}(5,5)$ under which coordinates of the SO(5,5) ExFT transform and which governs transformations of generators of the SO(5,5) exceptional Drinfeld algebra $(T_a, T^{ab}, T)$ with $a, b = 1, \ldots, 5$. The weight diagram for the spinorial representation $\mathbb{X}^M$ with $M = 1, \ldots, 16$ is given in Figure 7.

Now, from the diagram, it is clear that upon the first embedding, the geometric coordinates (equivalently, "physical" generators of the SO(5,5) EDA) correspond to the weights $(\mathbb{X}^1, \ldots, \mathbb{X}^5)$, while the rest correspond to winding modes. Upon the second embedding, the "physical" subalgebra of EDA is spanned by generators corresponding to the weights $(\mathbb{X}^{12}, \ldots, \mathbb{X}^{15})$. One notices that the two sets of physical coordinates do not intersect and one is able to perform such an SO(5,5) transformation as to shift all 1-forms $\sigma^a$ into the non-geometric set. Equivalently, this demonstrates the existence of two possible choices of the "physical" subalgebra inside the exceptional Drinfeld algebra with SO(5,5) symmetry, which do not conflict.

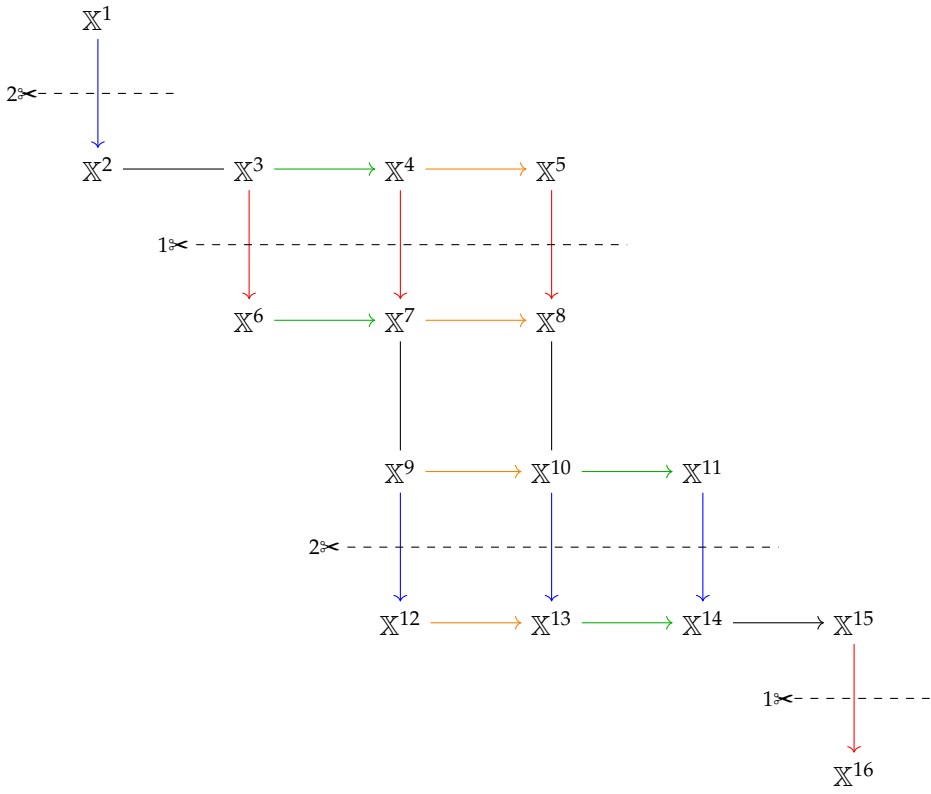

**Figure 7.** Weight diagram of the **16** of $\mathfrak{so}(5,5)$ with two possible embeddings of the $\mathfrak{gl}(5)$ subalgebra.

## 5. Discussion

In this work, a generalisation of the non-abelian T-duality Buscher rules for 10D supergravity backgrounds to 11D backgrounds was proposed. For that, one starts with the representation of the conventional NATD as a B-shift of the undressed generalised metric linearly proportional to dual coordinates $\Delta B_{ab} = f_{ab}{}^c \tilde{y}_c$ with further abelian T-dualities along all directions to turn all $\tilde{y}_a$ into geometric coordinates. Naturally, this translates into a procedure that starts with the C-shift of the generalised metric of exceptional field theory $\Delta C_{abc} = -3\tilde{y}_{d[a}f_{bc]}{}^d$, which produces a field configuration depending on dual coordinates. To end up with a solution of the supergravity equations, one either performs a formal conventional U-duality transformation that turns dual coordinates into geometric or chooses appropriate IIB section. These procedures can be understood as the construction of a background with a flat metric and gauge fields linearly depending on dual coordinates such that it has precisely the same generalised fluxes as the initial one. Such a defined background is then guaranteed to solve equations of double (exceptional) field theory and, hence, of the usual supergravity upon T(U)-duality of all winding directions. For the NAUD case, the procedure was checked to work only for backgrounds with a unimodular symmetry group, i.e., group manifolds with $f_{ab}{}^b = 0$. For NATD, such backgrounds would generate non-vanishing trombone gauging, which in general would require a generalised supergravity framework. A similar observation can be made in the ExFT case. Indeed, the tension between generalised flux components of the SL(5) theory due to terms containing $f_{ab}{}^b$ can be removed by passing a linear dependence on dual coordinates to the field $\phi$ proportional to the determinant of the external metric. At the level of fluxes, this has many similarities with the dilaton $d$ of DFT, whose linear dependence on dual coordinates gives rise to generalised supergravity. However, it is widely accepted that generalised supergravity does not exist in 11 dimensions based on the observation that no Weyl symmetry to break is present for the membrane. Observations made in the present work suggest further investigation of the possible relaxation of this statement.

From the algebraic point of view, the set of T-dualities along all directions is equivalent to replacing generators $T_a$ by their duals $\tilde{T}^a$ in the double Drinfeld algebra (Manin triple). For supergravity backgrounds, that means that one embeds the "physical" $\mathfrak{gl}(d)$ in two different ways: such that $T_a$ or $\tilde{T}^a$ transform in the fundamental **d** of $\mathfrak{gl}(d)$. This corresponds to the external automorphism of the $\mathfrak{o}(d,d)$ algebra replacing the deletion of one of the roots on either ends by the deletion of the root on the opposite end. This observation and the requirement for U-dualisation of all dual coordinates suggests understanding NAUD transformation as a switch between two "physical" algebras $\mathfrak{gl}(d)$ by the external automorphism of the corresponding exceptional symmetry algebra. We showed that for the algebra $\mathfrak{sl}(5)$, such a procedure can generate solutions of the conventional supergravity only when a spectator field is present, which is consistent with the observation made in [33]. Investigating the example of the algebra $\mathfrak{so}(5,5)$, one concludes that larger U-duality symmetry groups allow such non-abelian U-dualisation, and a solution of the equations of 11-dimensional supergravity can be constructed. The investigation of explicit examples based on the SO(5,5) and E$_6$ exceptional Drinfeld algebra is reserved for future work.

One becomes naturally interested in a generalisation of the obtained results to exceptional field theories to general manifolds with isometries along the lines of [30,31,35]. In this case, symmetries manifest themselves in the algebra of Killing vectors, which can be used to organise a tri-vector shift, in contrast to the 3-form shift in the present paper [62,63]. This provides tri-vector deformations of 11-dimensional backgrounds, which in certain cases follow the same scheme as in Figure 2, e.g., one considers the tri-vector deformation of Minkowski space–time, which in the IIB frame is again a Minkowski space–time, while solving the equations of generalised supergravity in the IIA frame [62]. A more detailed analysis of the relations between deformations and non-abelian dualities is required.

**Funding:** This research was funded by Foundation for the Advancement of Theoretical Physics and Mathematics «BASIS» and by Russian Ministry of education and science.

**Acknowledgments:** The author is thankful for the vivid discussions with I. Bakhmatov, K. Gubarev, E. Malek, and N. Sadik Deger, which motivated this project. The author thanks Yuho Sakatani for useful comments and suggestions. This work was supported by the Foundation for the Advancement of Theoretical Physics and Mathematics «BASIS» and by Russian Ministry of education and science. In part, the work was funded by the Russian Government program of the competitive growth of Kazan Federal University.

**Conflicts of Interest:** The author declares no conflict of interest.

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
