# Peer review of "On the Non-Abelian U-Duality of 11D Backgrounds"

_universe, doi:10.3390/universe8050276_

Round 1

Reviewer 1 Report

The author proposes how to generalise the procedure of non-abelian T-duality to an 11-d non-abelian U-duality, and compares this with the proposal put forward by the Exceptional Drinfeld Algebra. The author argues that when considering these non-abelian U-dualities on a 4-d space, they always involve a spectator field, i.e. a U(1) isometry and so reduce to just a non-abelian T-duality. The author also argues this matches the perspective from the Exceptional Drinfeld Algebra.

There are various assumptions hidden, which could be spelt out better. Moreover, the main results could be more clearly presented in the introduction, where there is no mention of the spectator fields, and "spectator fields" could be defined in the abstract.

However, I think for Universe, this suffices. 

Author Response

Dear referee,

many thanks for the comments on the text. The following corrections have been made.

  1. English language and style have been edited, including corrected misprints.
  2. In the lines 88-100 more clear explanation of the results of the paper has been given. In particular, it is emphasised, that the procedure guarantees to end up with a solution to supergravity equations
  3. Mention of spectator field in the abstract is rephrased in terms of isometry algebras and sounds now more correct.

Reviewer 2 Report

The author proposes a generalization of the non-abelian T-duality procedure for 10D supergravity to 11D supergravity. For that purpose, 7+10 exceptional field theory is used, which has both, 10D and 11D reductions. It is a timely contribution to the field and certainly uintersting for experts.

The paper is articulated in three main sections (2, 3, and 4) containing a review of non-abelian T-duality, the generalization to excpetional field theory, i.e. non-abelian U-duality, and a discussion of algebraic aspects, respectively. 

I can recommend the manuscript for publication, but I invite the author to define abbreviations such as NAT, NATD, PLTD, EDA, ExFT, DFT, NAUD.

Moreover, he should carefully look for spelling mistakes and grammar errors. 
For example, 
"no longer present" -> "are no longer present" before (10)
"oin" -> "in " line 170
"changing turing" line 222 ?
indices in (23)?
a_3 instead of a_2  in (28) ? 

and many more.

Author Response

Dear referee,

many thanks for the comments on the text. The following corrections have been made.

  1. English language and style have been edited, including corrected misprints in the text and formulas.
  2. All abbreviations have been appropriately defined.